# Short Communication: A simple workflow for robust low-cost UAV-derived change detection without ground control points

Kristen L. Cook[1], Michael Dietze[1]

[1]GFZ German Research Centre for Geosciences, Telegrafenberg, Potsdam, Germany

*Correspondence to*: Kristen L. Cook (klcook@gfz-potsdam.de)

**Abstract.** High quality 3D point clouds generated from repeat camera-equipped unmanned aerial vehicle (UAV) surveys are increasingly being used to investigate landscape changes and geomorphic processes. Point cloud quality can be expressed as

accuracy in a comparative (i.e., from survey to survey) and absolute (between survey and an external reference system) sense. Here we present a simple workflow for calculating pairs or sets of point clouds with a high comparative accuracy, without the need for ground control points or a dGNSS equipped UAV. The method is based on the automated detection of common tie points in stable portions of the survey area. We demonstrate the efficacy of the new approach using a consumer-grade UAV in two contrasting landscapes: the coastal cliffs on the Island of Rügen, Germany, and the tectonically active

Daan River gorge in Taiwan. Compared to a standard approach using ground control points, our workflow results in a nearly identical distribution of measured changes. Compared to a standard approach without ground control, our workflow reduces the level of change detection from several meters to 10-15 cm. This approach enables robust change detection using UAVs in settings where ground control is not feasible.

## 1 Introduction

Camera-equipped unmanned aerial vehicles (UAVs) and Structure from Motion (SfM) methods are increasingly being utilized as a low-cost method to conduct repeat topographic surveys in order to measure geomorphic change (Fonstad et al., 2013; Eltner et al., 2016; Anderson et al., 2019). To obtain high quality 3D models using SfM, precisely located ground control points (GCPs) are typically used (James and Robson, 2014; Carrivick et al., 2016) to both georeference the model and to improve the calculation of camera interior parameters and camera positions and orientations. This requires either the

deployment of GCP targets prior to UAV flights or the identification of existing natural or artificial features that can be used as targets. In either case, the locations of the GCPs must be precisely measured, typically using a differential GNSS (dGNSS) or total station (James et al., 2017).

In the absence of GCPs, models can also be created using direct georeferencing, which requires GPS locations of the camera positions (Carbonneau and Dietrich, 2017). For highly accurate results, this relies on having very accurate camera locations,

typically by using a UAV equipped with dGNSS (Turner et al., 2013; Hugenholtz et al., 2016). Direct georeferencing performed using only the GPS positions recorded by consumer-grade drones can lead to models that contain a range of errors

and distortions (Carbonneau and Dietrich, 2016; James et al., 2017). Model errors can also be reduced by complementing nadir surveys with oblique images in a convergent geometry (James and Robson, 2014), but this is typically recommended in conjunction with GCPs or dGNSS based direct georeferencing. Peppa et al (2018) presented a method for automatically

generating pseudo-GCPs in stable areas using DEM curvature and openness, but this relies on using surface texture to estimate stability, which may not be reasonable in all settings. In addition, the generation of DEMs may not be appropriate for all terrain types, such as overhanging cliffs. Feurer and Vinatier (2018) introduce a method to process sets of archival aerial photographs in the same SfM block to achieve accurate change detection with only a small set of poorly constrained GCPs (accuracy ~20 m) for scaling and georeferencing.

When considering accuracy in relation to change detection, we distinguish two different types: the real accuracy of an individual model and the comparative accuracy of a pair of models. Real accuracy includes both relative and absolute accuracy, or the internal accuracy (distortion) of the model and the accuracy of the scaling and georeferencing of the model. We use the term comparative accuracy to describe the accuracy of the change measured between model pairs, or to what degree the models are consistent with each other. High real accuracy should lead to high comparative accuracy, and is the

most desirable outcome, but it may be possible to achieve high comparative accuracy for model pairs with low real accuracy. For example, if two models are subjected to the same incorrect transformation or rescaling, their real accuracy will be affected while their comparative accuracy remains unchanged.

While high real accuracy is desirable, some settings of interest for change detection preclude the deployment or measurement of GCPs, and dGNSS-equipped UAVs may be prohibitively expensive. Therefore, an alternative method for

achieving high comparative accuracy of survey pairs could open up new types of settings to event monitoring using low-cost UAVs. Here, we introduce a simple workflow involving the co-alignment of photographs from different surveys; our method is similar to that of Feurer and Vinatier (2018), but is generalized to any set of repeat SfM surveys and requires no GCPs. Using data from two contrasting study areas: a bedrock gorge in Taiwan and a steep cliff coast in northern Germany, we demonstrate that we can achieve high comparative survey accuracy and low limits of change detection using a low-cost off

the shelf UAV without ground control points. Our workflow is extremely simple, can be performed entirely with the software Agisoft Photoscan Pro (now called Metashape Pro), and could be made fully automated.

## 2 Study area

We first present data from the Daan River, a bedrock gorge in Taiwan. In this system, the river experiences large changes between survey periods, while the surrounding area has variable degrees of vegetation cover and remains stable aside from

60 vegetation growth. The gorge also experiences localized erosion of its steep to vertical walls. An extensive description and analysis of survey accuracy in this setting can be found in Cook (2017), who estimates a level of detection of 10-30 cm (depending on surface characteristics) for GCP-constrained surveys. Because we have ground control information for these surveys, we can compare GCP-constrained changes to changes measured using our workflow without GCPs.

The primary study area is located in Jasmund National Park on the island of Rügen, Germany, where steep to overhanging coastal cliffs up to 118 m high are eroding rapidly (Schulz, 1998) (Figure 1). Our study area comprises about 7 km of coastline, from the Königsstuhl in the north to the town of Sassnitz in the south. The cliffs, composed of chalk and glacial till, experience frequent rockfalls and collapses during the winter months. During our study period from 2017-2019, these failures varied in size from a few $m^3$ to about 4000 $m^3$. While rockfalls are relatively common, they affect a small proportion of the total cliff area, and the rest of the cliff face remains stable, with no discernable internal deformation.

This cliff coast presents a challenging environment for UAV-based surveying. The cliff sections are out of bounds, access to the base of the cliffs is limited and can be dangerous, the forest above the cliffs limits both ground visibility and the communication range of the UAV, and strong winds are common. In addition, the coast is a long linear feature that precludes complicated flight patterns, and flying close to the cliff is restricted to protect peregrine falcons nesting there. However, because cliff collapses can represent a significant hazard to National Park visitors, there is a strong interest in a rapid and easy to implement method of monitoring cliff activity. This combination of characteristics makes the cliff a good location for demonstrating the applicability of our workflow, as it is a setting in which conventional methods are unsatisfactory.

# 3 Methods

## 3.1 Data acquisition

Daan River surveys were flown with a Phantom 3 Advanced UAV using flight planning software, yielding grids of nadir photographs from 35-60 m above ground level. Here, we marked ground control points with spray paint and measured their locations using a dGPS with 1-2 cm accuracy. We compare subsets of surveys conducted in May 2017 and Jan 2018, which used 14 and 12 ground control points and 197 and 298 photographs, respectively.

Rügen surveys were conducted by manually flying a DJI Mavic Pro UAV from three to seven locations along the top of the cliff (depending on wind conditions and the impact of foliage on the UAV communication range). Photos were taken every 3 seconds, and typically two passes were made for each cliff section – one at lower altitude with the camera more oblique, and one at higher altitude with the camera more nadir (Figure 1). Typically, the camera pitch was 40 to 80 degrees from nadir and flight elevations ranged from 30 to 150 m above sea level, depending on the height of the cliff. In order to ensure adequate coverage, the UAV was positioned so that each photo included the full vertical extent of the cliff. As a result, the distance between the UAV and the cliff varied depending on the cliff height. Flight heights and distances from the cliff also had to be adjusted to weather conditions such as wind speed and sun glare. Each flight took 20-30 minutes, so the full 7 km stretch of cliff could be surveyed in a few hours. Each survey contained 1000-2000 photographs. We also conducted several partial surveys that covered smaller segments of the cliff coast during the winter of 2017-2018. We have no ground control points for the surveys. The base of the cliff can only be accessed in a few locations, and National Park regulations prohibit employees or associates from working along the cliff base. Deploying ground control points only on the cliff top would result in a linear array of points, a geometry that can lead to large errors.

*3.2 Data Processing*

SfM processing was done using Agisoft Photoscan Pro (v. 1.4.2). In order to decrease processing time, the 7 km long Rügen study area was separated into five overlapping segments. In this paper, we will show data from just two of these segments –

100 the Kieler Bach and Königsstuhl sections.

As a control, we processed the data using a standard Agisoft workflow in which each survey is processed separately. For the Daan example, we used the GCP information to georeference each survey. For the Rügen surveys, the only georeferencing information was provided by the photo GPS tags created by the DJI Mavic Pro or Phantom 3. Because the elevation data reported by these UAVs often contain systematic offsets, we used the known elevations of the launch points to correct the

105 elevations of the cameras for each flight. Photos were aligned (using high quality and 40,000 and 4000 key and tie point limits, respectively), tie points with reconstruction uncertainty greater than 50 were removed, and the alignment was optimized (using adaptive camera model fitting). Dense clouds were calculated using medium quality and aggressive depth filtering, exported into CloudCompare (CloudCompare 2.10.1, 2019), and co-registered using iterative closest point fitting. Then the M3C2 algorithm (Lague et al., 2013) was used to compare point clouds from successive surveys, using a projection

diameter of 0.5 m, normal scales from 0.5 m to 4.5 m by 1 m steps, and core point spacing of 0.25 m. We then trimmed areas of vegetation using standard deviation and point density filters (Cook, 2017).

We then tested a workflow, which we term co-alignment, that involves processing survey pairs together (Figure 2). To do this, we imported the photographs from two different surveys into a single chunk in Photoscan and performed the point detection and matching, initial bundle adjustment, and optimization steps on the combined set of photographs, using the

115 same parameters as above. We created different camera calibration groups for each survey, so the calculated camera calibration parameters can differ between surveys. If there is sufficient similarity in the photographs between the two survey periods, key points can be matched between photos from different surveys and common tie points will be generated. After the alignment and optimization steps were finished, we separated the photos from the different surveys by creating two duplicates of the original chunk and removing photos as needed, thus preserving the sparse clouds, position information, and

120 the camera calibrations. We then calculated dense clouds for each survey period and compared the resulting point clouds using M3C2 in CloudCompare, with the same parameters listed above.

## 4 Results and discussion

The Daan River surveys enable us to compare the effectiveness of the co-alignment workflow without GCPs to a traditional workflow using GCPs. We find that the co-alignment workflow results in a change map and density curve that are almost

125 identical to those produced using the GCPs (Figure 3). The only apparent differences between the two change maps occur on the upper edge of the area, where the photograph coverage becomes marginal and errors occur in both the GCP-constrained

and co-aligned comparisons. This provides evidence that co-alignment can be used for change detection with a level of detection comparable to that of a survey grade GCP-constrained pair of models.

For the Rügen data, we assessed the comparative accuracy of the resulting model pairs based on the measured change in stable areas of the cliff. Areas of poor fit can be distinguished from areas of real change by the spatial pattern of the differences, the sharpness of the boundary, and by visual inspection of the before and after photographs (Figure 1, 4-6).

Using the standard workflow, the point clouds from successive surveys each contain distinct errors and distortions. Because the error in each cloud is independent of the other cloud, the point clouds are distorted relative to each other and typically cannot be co-registered well, resulting in large errors in the change detection. The error varies throughout the model area, depending on the distortion of the individual models. The spatial pattern of error will also depend on the method used to co-register the two point clouds. For the example shown in Figure 4, erroneous changes of up to 5 m are measured on the edges of the models and of up to 2.5 m in the center. Throughout the model area, up to 1-2 meters of change are erroneously detected in many stable areas, indicating that real changes of this magnitude would be below the level of detection. For the Rügen study area, this level of detection would preclude the use of UAV surveys to monitor small cliff failures.

When the cameras from multiple surveys are co-aligned, the resulting point clouds still contain distortions, but if the procedure is successful, they have been fit to a common geometry and the distortions are consistent between the models. As a result, these errors do not influence comparisons between the models, comparative accuracy is much higher and robust change detection can be performed. We find that the measured change in stable areas is substantially less than in the control case, and therefore smaller amounts of real change can be detected (Figure 4). For the examples shown here, the level of detection has been reduced from several meters to as low as 15-20 cm. Small cliff failures, bands of more diffusive erosion at the base of the cliff, and even the growth of individual bushes can be reliably detected (Figures 4-6).

The increase in comparative accuracy is due to the generation of tie points between photographs from different surveys. These tie points, if they are well distributed, enforce a common geometry between the different surveys. We can evaluate the number of common tie points between surveys by comparing the number of points in each sparse cloud following chunk duplication and photo removal (Figure 2) to the number of points in the sparse cloud generated during the combined alignment. Tie points generated using only photos from survey 1 will be removed when the photos from survey 1 are removed, while tie points generated using photos from both surveys will remain. If common tie points were generated, the two separated sparse clouds have more total points than the original, with the difference being the number of common points (Table 1). Note that this is distinct from the number of matches, as each tie point may be used in multiple matches.

Even when relatively few common tie points are generated, or when they are irregularly distributed, a successful alignment can be achieved. For example, Figure 5 shows a section of the Rügen study area that is heavily vegetated, with only isolated patches of bare cliffs. While no common tie points can be generated in the vegetated areas, as long as there are common tie points distributed throughout the cliff sections, a relatively good comparative accuracy can be achieved, as illustrated for April 2018 – May 2018 (Figure 5B). However, if there are sections of the cliff where no matches can be made, then large comparative errors can result, as is shown in Figure 5C for the survey pair Oct. 2017 – April 2018. This survey pair had both

a low number (1355) and percentage (0.4%) of common tie points compared to the April 2018 – May 2018 pair, which had 3402, or 1% common tie points. More importantly, there were no common tie points generated in a ~350 m long stretch at one end of the model, leading to up to 1.5 m of comparative error in this section of the cliff. This illustrates that if common tie points are not distributed through the full extent of the model, edges of the models may not align well. The Daan River example further demonstrates that the distribution of tie points is more important than their number, as good alignment was achieved throughout most of the model despite the generation of only 900 common tie points (0.3% of the total).

## 4.1 Potential limitations

In order to get a successful alignment, tie points linking the photos from different surveys must be detected, and false matches must be avoided. If the appearance of the area changes too much between surveys or if too much of the area of interest has changed, sufficient tie points may not be generated, as described above. Therefore, well-distributed stable areas with a consistent appearance are required for successful alignment. In the examples presented here, we did not observe any false matches, as surface changes were always accompanied by changes in appearance, preventing the detection of matches in unstable areas. In settings with large-scale surface deformation, such as a slow moving landslide or deep seated gravitational slope deformation, this may not be the case, and it is possible that points may be matched in unstable areas. In such settings, care should be taken to evaluate the reliability of the common tie points.

For a single pair of surveys, the co-alignment workflow has a limited impact on processing time. Due to non-linear scaling between the number of photos and the processing time, performing the point matching and camera alignment step once with n photos will take longer than performing it twice with n/2 photos, but this effect will be relatively minor until the number of photos becomes large. The more significant impact on processing time comes from the requirement that for each survey set to be compared, the entire chain of processing from point matching to dense cloud construction must be redone. This can greatly increase the total processing time for large sets of surveys. For example, for a set of four surveys, A, B, C, and D, a series of pairwise processing and comparison (A-B, B-C, C-D) would require the point matching and camera alignment step to be performed three times and would require the construction of six dense clouds (surveys B and C would each have two dense clouds). This processing time can be reduced by applying the method to larger sets of surveys. We have simultaneously co-aligned photographs from up to 4 different epochs to obtain a set of mutually comparable point clouds from 2017-2018 (Figure 6). In some cases, an unsuccessful alignment of two surveys can be improved by adding a third survey. For example, if changes in surface appearance (lighting, shadows) or in camera obliquity prevent the detection of sufficient common tie points between the original two surveys, a third survey that generates enough common tie points with each of the original two can lead to successful alignment of all three surveys. However, despite the possibilities for batch processing, the fundamental drawback of this method is that it does not result in a definitive model for a given survey period - models that were constructed based on co-alignment of one set of surveys cannot be re-used for comparison to an additional survey.

While this procedure can yield point clouds that are well-aligned relative to each other and can be robustly compared, the real accuracy of the point clouds is not enhanced. The point clouds still contain errors and distortions, and measurements of distance, area, or volume should be interpreted accordingly. In the Daan river case, the point clouds generated without GCPs had a typical doming distortion, with up to 0.75 m of error on the edges of the model (relative to the GCP-constrained cloud). Thus, where ground control is feasible to obtain, GCP-constrained georeferencing is preferable to the co-alignment workflow if accuracy on the order of cm or better is desired. The combination of co-alignment and GCPs used by Feurer and Vinatier (2018) demonstrates a potential way forward to efficiently obtain both high real and comparative accuracy. If GCPs can be deployed and measured for just one survey, they can be used in conjunction with the co-alignment workflow to refine the model geometry for additional surveys. This could lead to improved real accuracy for all models while significantly reducing the field time needed for repeat surveys.

## 5 Conclusions

We show that for environments such as coastal cliffs where the use of ground control points is not possible or not feasible, UAV-based change detection can still be performed with a high degree of confidence if there is sufficient stable area between successive surveys. The workflow we present is quite simple and involves performing image matching and bundle adjustment simultaneously on photographs from pairs or sets of different surveys. This technique may be particularly useful for monitoring processes such as rockfalls, which typically involve steep settings that are difficult to access and exhibit discrete regions of change set within large stable areas.

**Author contribution**. Both authors carried out the field campaigns and designed the project. KLC did the data analysis and wrote the manuscript with input from MD.

**Competing interests**. The authors declare that they have no conflict of interest.

**Acknowledgements**. We thank the Jasmund National Park staff, in particular Stefanie Puffpaff and Ingolf Stodian, for their support of the project and assistance in the field. We also thank Benjamin Huxol and Oliver Rach for logistic support and flight involvement. Niels Hovius is thanked for financial and infrastructure support of the Jasmund observatory and the drone surveys.

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

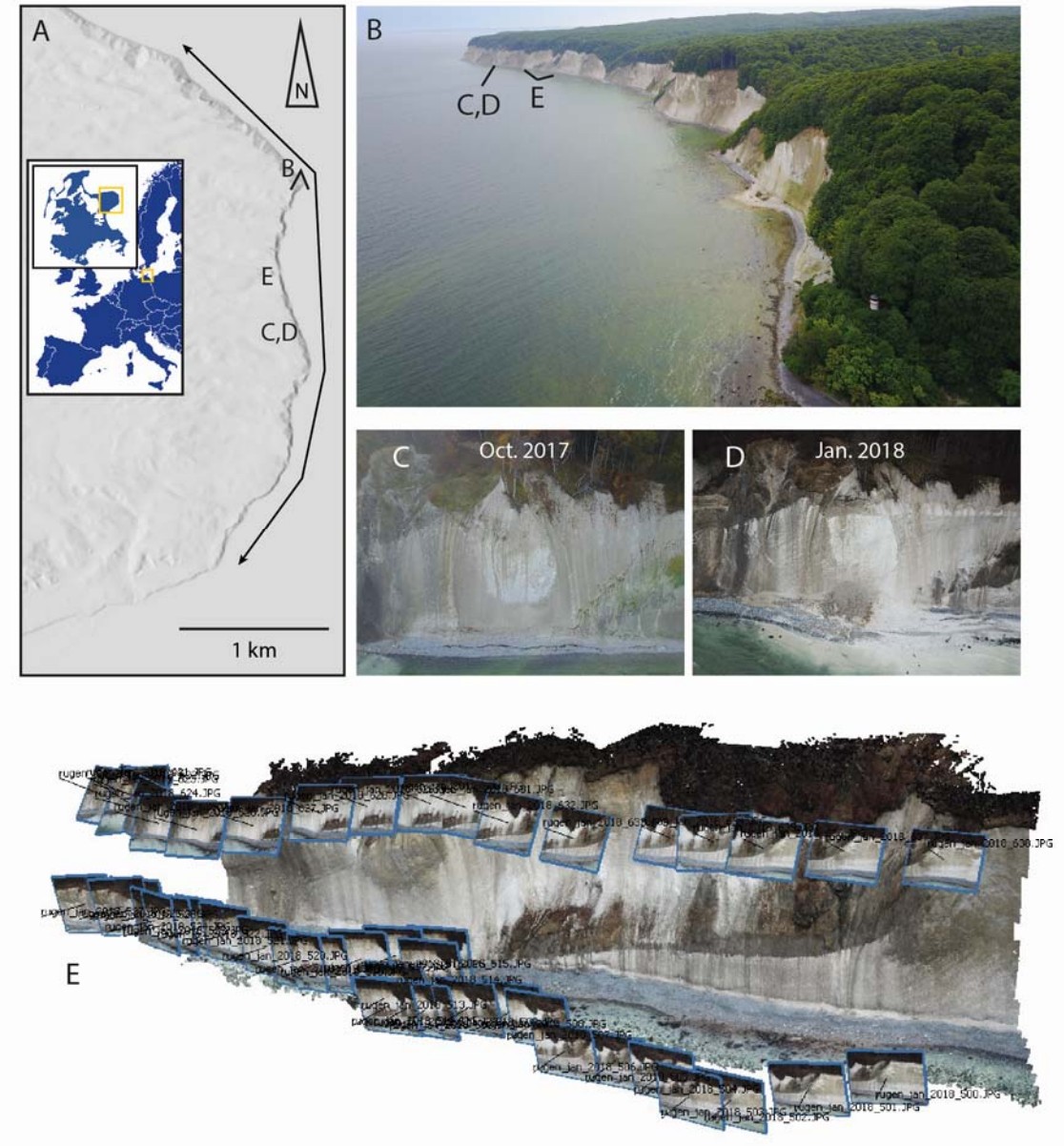

**Figure 1: A) Location of the Rügen study area, black line shows the studied coast section and the locations of panels B-E are indicated. B) Photo of the cliff coast in May 2018, view looking south. C) and D) before and after photos of a cliff failure. E) Example of survey geometry, with two passes at different altitudes and camera orientations, from Jan. 2018.**

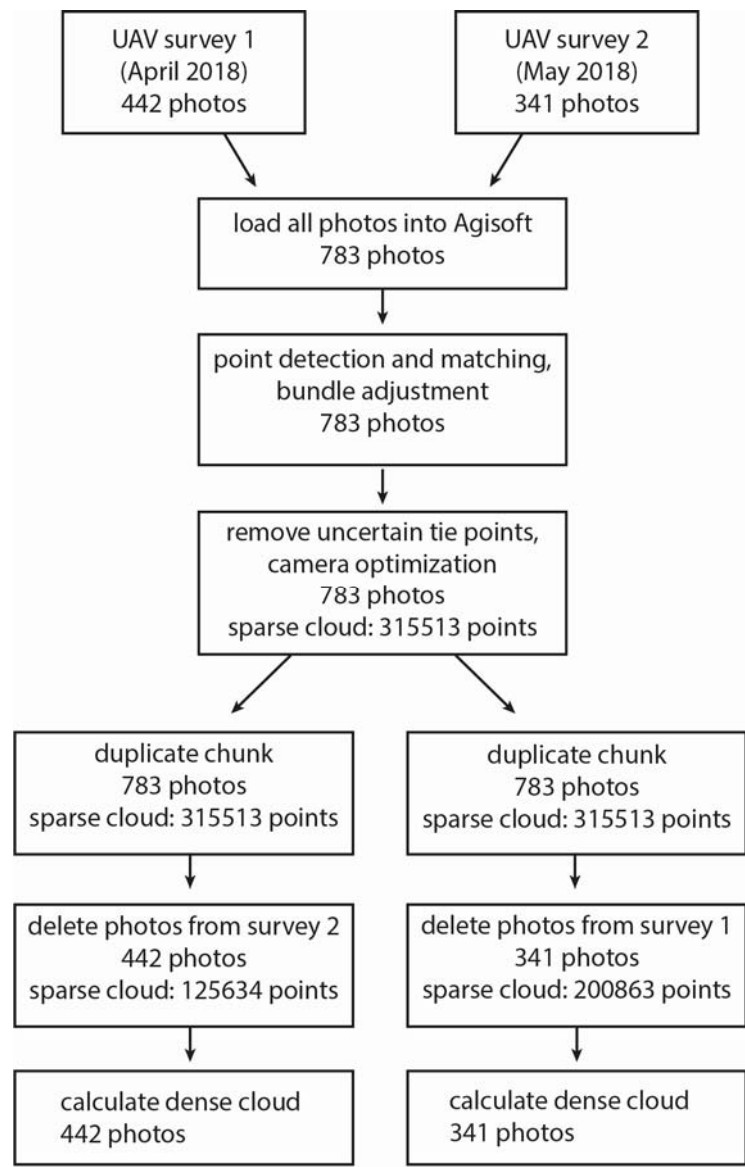

**Figure 2: Workflow of the co-alignment processing method, with numbers from the April-May 2018 Rügen comparison for reference.**

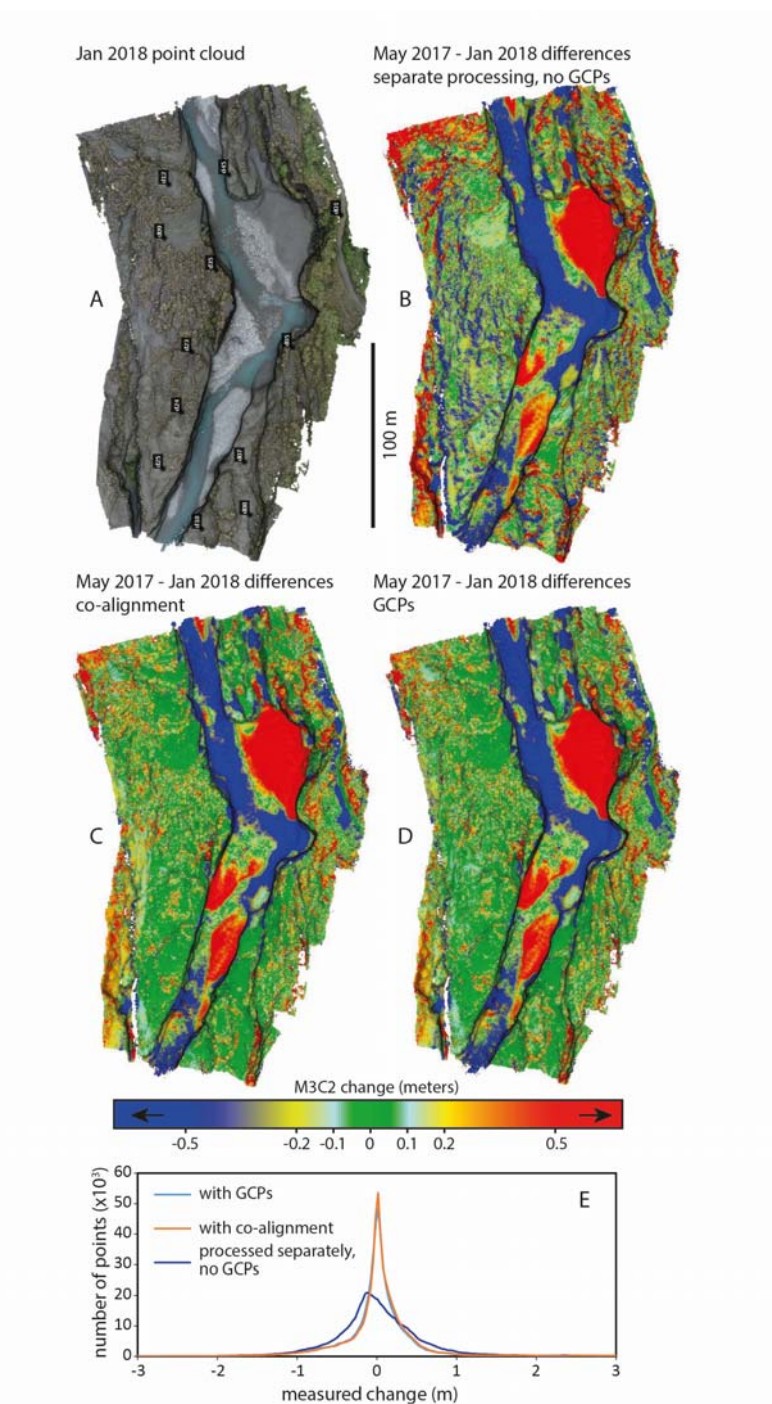

Jan 2018 point cloud

May 2017 - Jan 2018 differences
separate processing, no GCPs

100 m

A

B

May 2017 - Jan 2018 differences
co-alignment

May 2017 - Jan 2018 differences
GCPs

C

D

M3C2 change (meters)

-0.5    -0.2 -0.1  0  0.1 0.2       0.5

number of points (x10³)

with GCPs

with co-alignment

processed separately,
no GCPs

E

measured change (m)

**275**    **Figure 3: Daan River comparisons. A) Jan. 2018 point cloud with the ground control points shown. B) M3C2 differences between May 2017 and Jan. 2018 point clouds processed separately with no GCPs. C) M3C2 differences between May 2017 and Jan. 2018 point clouds processed using the co-alignment workflow. D) M3C2 differences between May 2017 and Jan. 2018 point clouds processed separately using GCPs. E) Density curves of the measured changes shown in B, C, and D.**

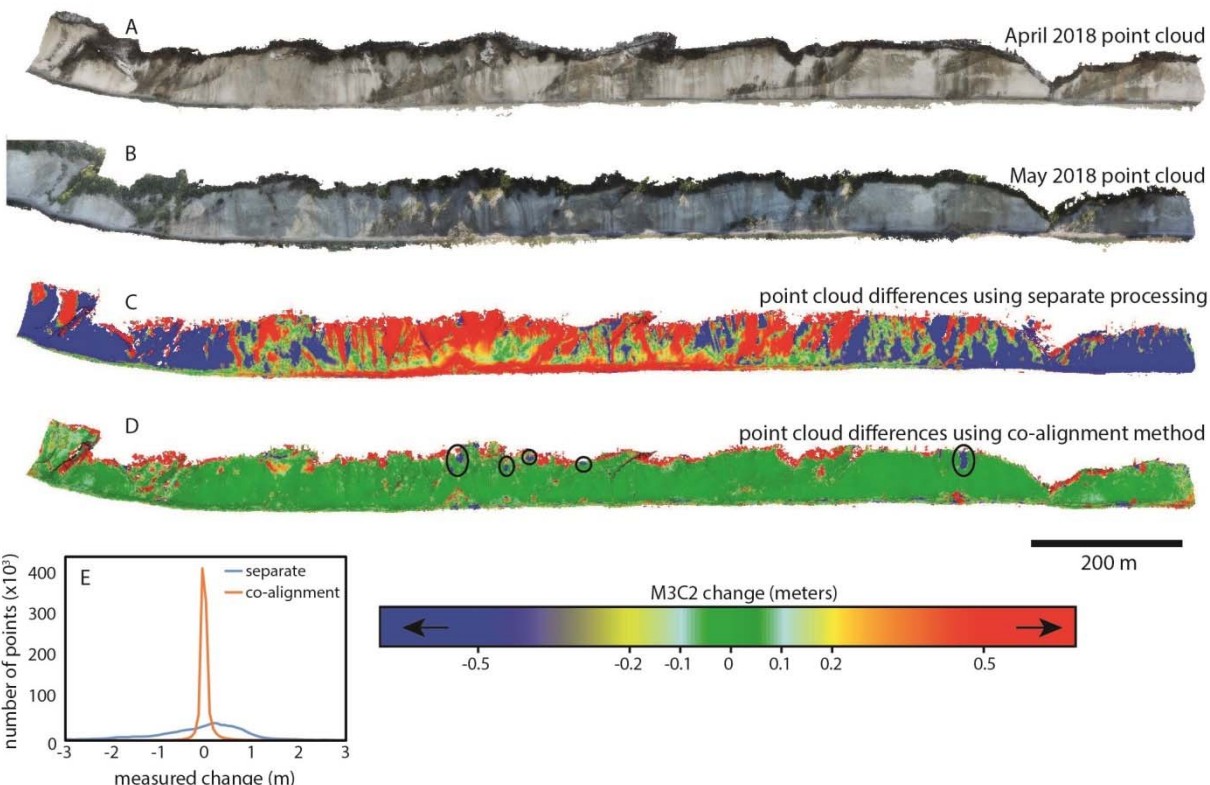

**Figure 4: Cloud-cloud differences between the April 2018 and May 2018 surveys in the Kieler Bach section of the coast, calculated using the M3C2 algorithm. A) April 2018 point cloud. B) May 2018 point cloud. C) M3C2 differences between point clouds created using the standard workflow. D) M3C2 differences between point clouds created using the co-alignment workflow. High values of positive change at the top of the cliff are due to leaf growth on the trees. Isolated sections of positive change on the cliff face are also related to growth of bushes and trees. In panel D, several small failure events can be identified on the cliff face (circled). These have been confirmed visually using the before and after photographs. E) Density curves of the measured changes shown in C and D.**

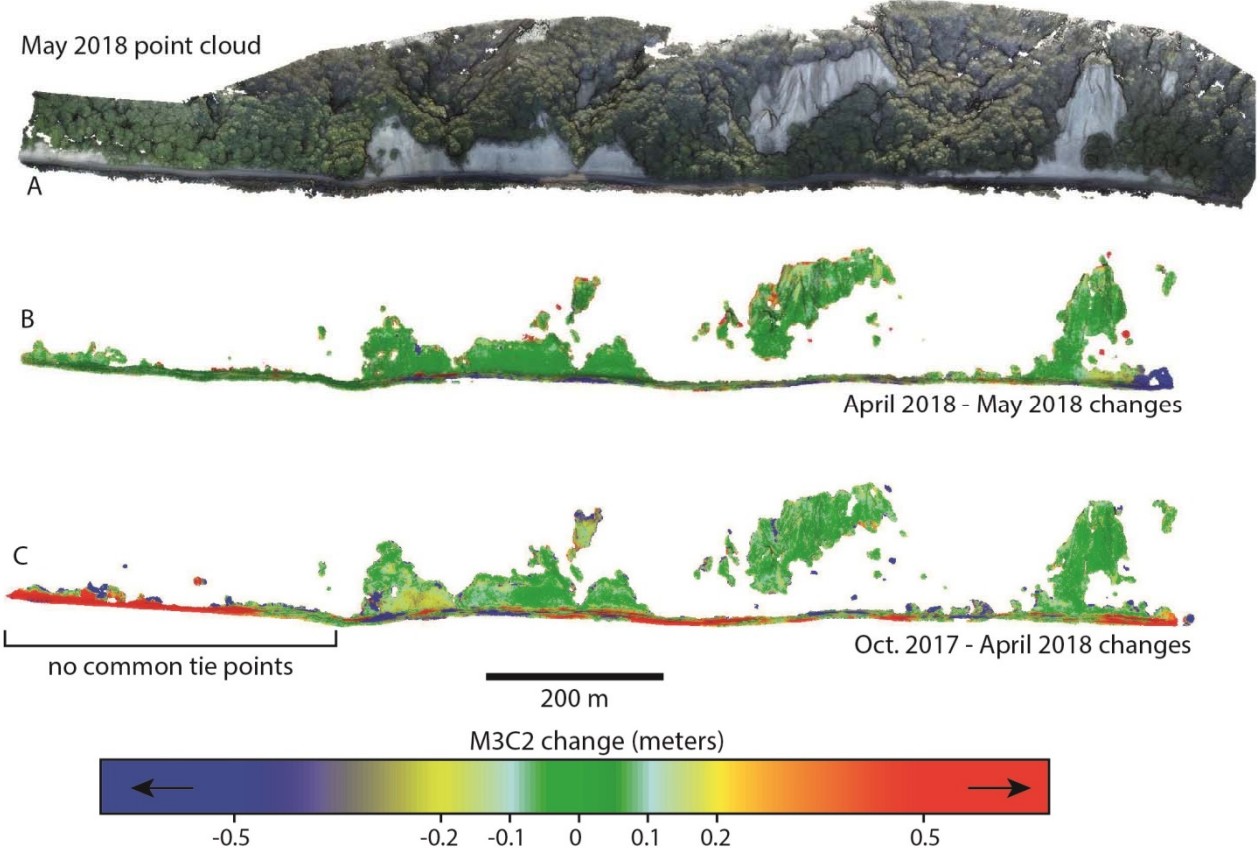

May 2018 point cloud

A

B

April 2018 – May 2018 changes

C

no common tie points

200 m

M3C2 change (meters)

-0.5        -0.2  -0.1   0   0.1  0.2        0.5

Oct. 2017 – April 2018 changes

**Figure 5: Cloud-cloud differences in the heavily vegetated Königsstuhl section of the coast. A) May 2018 point cloud showing the extent of the vegetation. B) M3C2 differences between April 2018 and May 2018 point clouds. The vegetation has been removed using standard deviation and point density filters. Leaf growth results in very high measured changes in the vegetated areas, so only the bedrock cliff sections are shown. C) M3C2 differences between Oct. 2017 and April 2018 point clouds. A lack of common tie points detected in the left side of the region results in relative distortion of the models and high errors in the change detection.**

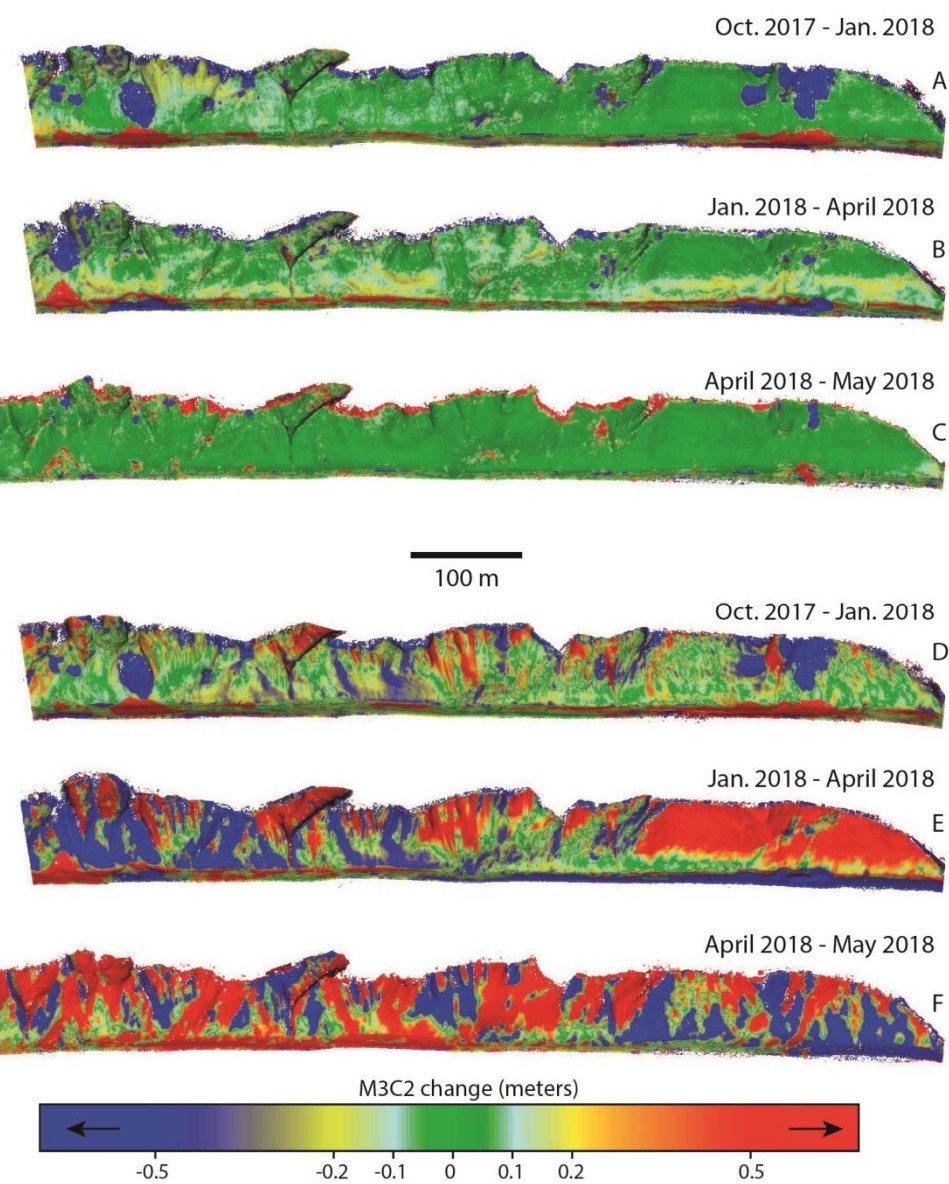

Figure 6: Changes calculated from batch co-alignment of four surveys simultaneously. A-C) M3C2 changes between successive surveys following simultaneous co-alignment. In panel B) bands of change in the lower half of the cliff show more diffuse erosion due to mechanical weathering. D-F) the same comparisons following the separate processing workflow.

| Survey | UAV | Number of photographs | Sparse cloud points | Common tie points | % Common tie points |
|---|---|---|---|---|---|
| **Daan River (figure 3)** | | | | | |
| May 17, 2017 | Phantom 3 Adv. | 197 | 136479 | | |
| Jan. 30, 2018 | Phantom 3 Adv. | 298 | 168953 | | |
| Combined alignment | | | 304532 | 900 | 0.30 |
| | | | | | |
| **Rügen Kieler Bach (figure 4)** | | | | | |
| April 03, 2018 | Mavic Pro | 442 | 125634 | | |
| May 29, 2018 | Mavic Pro | 331 | 200863 | | |
| Combined alignment | | | 313513 | 12984 | 4.14 |
| | | | | | |
| **Rügen Konigsstuhl (figure 5)** | | | | | |
| April 03, 2018 | Mavic Pro | 250 | 111464 | | |
| May 29, 2018 | Mavic Pro | 249 | 157677 | | |
| Combined alignment | | | 264597 | 4544 | 1.72 |
| | | | | | |
| Oct. 18, 2017 | Mavic Pro | 414 | 195901 | | |
| April 03, 2018 | Mavic Pro | 246 | 117227 | | |
| Combined alignment | | | 311773 | 1355 | 0.43 |
| | | | | | |
| **Rügen batch processing (figure 6)** | | | | | |
| Oct. 18, 2017 | Mavic Pro | 839 | 363485 | | |
| Jan 24, 2018 | Mavic Pro | 338 | 125121 | | |
| April 03, 2018 | Mavic Pro | 442 | 128640 | | |
| May 29, 2018 | Mavic Pro | 575 | 324391 | | |
| Combined alignment | | | 918741 | 22896 | 2.49 |

**Table 1: survey characteristics**