# Peer review of "Short Communication: A simple workflow for robust low-cost UAV-derived change detection without ground control points"

_Earth Surface Dynamics, 2019_

## Referee Comment (RC1) · Anonymous Referee #1 · 17 Jun 2019

The short communication introduces a workflow to process multi-temporal data for accurate change detection although no GCPs are available. Thereby, images from multiple survey campaigns are processed at once. Afterwards, the orientated images of the individual surveys are split to retrieve the corresponding point clouds of each campaign for change detection. The idea is simple but very effective. The manuscript is well-structured and easy to follow. The results are presented comprehensively and support the introduced method. However, some issues remain regarding the explanation of the approach (especially terminology) that should be addressed in a revised manuscript. Furthermore, the authors should consider the Time-SIFT publication by Feurer & Vinatier (2018) because it describes a similar approach more detailed for

applications to archival imagery. Please, see below for some detailed comments.

P1L23: It is not clear what the authors refer to with camera optical parameters. Are these the interior orientation parameters. If yes, it should be mentioned that the GCPs are also used to refine the parameters of the exterior orientation besides the interior parameters.

P1L29: It might be better to refer to dGNSS instead of dGPS as also other satellites can be used for geo-referencing.

P1L31: Dietrich, 2017

P1L31: Model errors can also be reduced. . .

P2L33-35: Are these control points tie points or ground control points? If they are GCPs, where does the reliable/accurate 3D information come from? And if they are tie points, I would avoid the term control points.

P3L65-71: This paragraph seems to be a little bit off-topic if it is left as it is. A better explanation why these challenges are displayed should be provided. For instance, why is the changing appearance of the cliff relevant? Does that potentially impact feature detection and matching? Furthermore, a final statement might improve that paragraph, as well, highlighting that this study at the cliff is a very suitable study to demonstrate the usability/necessity/benefits of the authors' approach. Although, this intention is probably meant in the paragraph it might be suitable to mention this explicitly.

P3 chapter Methods: I would suggest to include sub-headings for data acquisition and co-alignment processing to improve the readability.

P3L75-76: Did the authors also consider check points as an independent reference of the reconstruction accuracy? With 14 and 12 GCPs this should be possible.

P3L94: I thought, only the Mavic Pro was used for data acquisition (but also a Phantom is mentioned here)?

P4L97: What is the unit for the reconstruction uncertainty? According to Agisoft, the reconstruction uncertainty somehow relates to the base-height ratio. But how is the reconstruction uncertainty calculated?

P4L98: How is the adaptive camera model fitting working? What is the difference to the approach without adaptive fitting?

P4L99: The fine registration in CloudCompare is done via ICP (iterative closest point) fitting. Maybe, it might be preferable to state the actual performed algorithm rather than the tool name.

P4L105: The alignment optimization is actually also a bundle adjustment, however considering some refined parameter settings and/or referencing information. Thus, this might be rephrased to avoid confusion of the reader.

P4L116-118: Is it possible to express these differences between both change maps in numbers, e.g. considering the average of deviations between both maps? This question would also be relevant for the Rügen analysis. Furthermore, did the authors also check accuracies at check points? They might be helpful to assess how well changes are detectable with the reference in general.

P4L124-125: However, this depends on how the models are aligned. If GCPs or stable areas are used, I am not certain if this statement still holds. Of course, if ICP is used than these distortions can lead to difficulties in the alignment (depending on how strong these distortions are).

P5L128-129: I am not sure if I understand that sentence correctly. Changes between 1 and 2 m are common at the observed cliff on Rügen? Thus, the noise in the data is higher than the common changes at the cliff?

L123-129: Maybe the entire paragraph can be rewritten to improve clarity regarding model related distortions and issues due to alignment approaches.

P5L130-131: Maybe it is worth to extent the explanation that the simultaneous alignment of all campaigns leads to the circumstance that the highly spatially correlated errors (James et al., 2017), which also depend on the image observations (i.e. tie points), are potentially situated at the same locations in the individual models (because image orientation across surveys are constrained to the same tie-points) and therefore mitigated during point cloud differencing.

P5L135: Figures 4 A-C

P5L153: I would not state that edges are the issue but rather areas outside the tie point region.

P6L160-163: Maybe this statement should be separated more clearly from the previous because another aspect is discussed. The first aspect is referring to too strong changes of the surface and therefore failing to find matches and the second refers to changes of the entire surface but remaining a general similar appearance and thus falsely retrieving matches.

P6L164: What do the authors refer to when they are talking about scaling between numbers of photos?

P6L164-168: I have a little bit difficulty to understand that sentence. Do the authors mean that with each new campaign all the campaigns have to be re-processed?

P6L168-169: Might it not be possible to only compare from one survey to the next to avoid increasing the processing time with each new survey, although this might be less favorable for error propagation? Maybe it might worth testing in a future study how well campaign to campaign processing performs compared to reprocessing everything.

P6L177-178: Maybe the combination of both is most suitable (e.g. as discussed by Feurer & Vinatier, 2018). Align all campaigns in one workflow (this might also improve general model accuracy as more image observations will be available) and scale/geo-reference the whole project with GCPs (from just one campaign).

References: Feurer, D., Vinatier, F. (2018): Joining multi-epoch archival aerial images

in a single SfM block allows 3-D change detection with almost exclusively image information. ISPRS Journal of Photogrammetry and Remote Sensing, 146, 495-506. James, M., Robson, S., Smith, M. (2017): 3-D uncertainty-based topographic change detection with structure-from-motion photogrammetry: precision maps for ground control and directly georeferenced surveys. ESPL, 42, 1769-1788

---

## Referee Comment (RC2) · Anonymous Referee #2 · 17 Jul 2019

This is an interesting study on the possibility of improving the comparative accuracy of multiple surveys by co-processing the image sets when stable areas can be found and matched in a particular area. Rather than the workflow itself (which is hardly a proper workflow but just a modification of the standard SfM pipeline), I found the greatest merit of this work drawing the attention to this co-alignment possibility, that in many cases my be discarded or overlooked and can help to improve the quality of the results. My main comments to this work are the following (please check the annotated pdf for specific comments throughout the manuscript): 1. I would strongly suggest to include in the manuscript title the main limitation of the workflow. i.e. the presence of stable areas. The authors have acknowledged this in the limitations and

conclusions sections and should be specified in the title since is a major requirement. 2. The authors are too focused on the geomorphological settings (cliffs, rivers and such), which is not bad, but a relevant part of the SfM community works on more artificial environments such as agricultural settings where hardly stable areas can be found. How applicable would be the co-aligment in these cases? A quick literature review could give the authors a general view of the types of scenarios in which the SfM approaches are being applied and maybe they could comment in more detail to what extent their method is feasible to be applied. 3. I recommend checking the comments on the annotated pdf. There are some inconsistencies in the structure (like not following the order in the study sites), lack of information in the figures and poorly structured information (like in Table 1). A correction of these formal aspects can produce an improvement of the manuscript.

Please also note the supplement to this comment:
https://www.earth-surf-dynam-discuss.net/esurf-2019-27/esurf-2019-27-RC2-supplement.pdf

**Supplement:**

[revised manuscript text omitted]

**Página: 2**

T Número: 1 Autor:     Asunto: Resaltado     Fecha: 17/07/2019 7:28:07
GCPs- or dGPS-based

T Número: 2 Autor:     Asunto: Resaltado     Fecha: 17/07/2019 7:28:28
al.

T Número: 3 Autor:     Asunto: Resaltado     Fecha: 17/07/2019 7:28:52
Consider using :

T Número: 4 Autor:     Asunto: Resaltado     Fecha: 17/07/2019 7:29:20
presenting?

T Número: 5 Autor:     Asunto: Resaltado     Fecha: 17/07/2019 7:31:01
1. Please justify the selection of these two study areas
2. Rephrase the sentence to be shorter and better structured

T Número: 6 Autor:     Asunto: Resaltado     Fecha: 17/07/2019 7:30:56
Include company and country or reference.

T Número: 7 Autor:     Asunto: Resaltado     Fecha: 17/07/2019 7:33:21
This can be removed.

T Número: 8 Autor:     Asunto: Resaltado     Fecha: 17/07/2019 7:38:04
No information of this study area has been included as a figure. Please provide a map and pictue if appropriate, similarly to the Rugen site.

T Número: 9 Autor:     Asunto: Resaltado     Fecha: 17/07/2019 7:33:21

Autor:     Asunto: Nota adhesiva          Fecha: 17/07/2019 7:33:40
Why not describing this first, being the primary area?

[revised manuscript text omitted]

Número: 1 Autor:     Asunto: Resaltado     Fecha: 17/07/2019 7:42:02
This was made after or before the fine registration and M3C2 algorithm?

Número: 2 Autor:     Asunto: Resaltado     Fecha: 17/07/2019 7:47:23
Can you provide the results when both surveys are processed independently (no co-alignment)?

[Figure]

align the two point clouds. For the example shown in Figure 4, erroneous changes of up to 5 m are measured on the edges of the models and of up to 2.5 m in the center. Areas with 1-2 m of measured change are common, so real changes on the order of a few meters will not be detected in this comparison.

130   When the cameras from multiple surveys are co-aligned, the resulting clouds still contain distortions, but if the procedure is successful, they have been fit to a common geometry, so comparative accuracy is much higher and robust change detection can be performed. We find that the measured change in stable areas is substantially less than in the control case, and therefore smaller amounts of real change can be detected (Figure 4). For the examples shown here, the level of detection has been reduced from several meters to as low as 15-20 cm. Small cliff failures, bands of more diffusive erosion at the base of

135   the cliff, and even the growth of individual bushes can be reliably detected
[Figure]
(Figures 4-6).

The increase in comparative accuracy is due to the generation of tie points between photographs from different surveys. These tie points, if they are well distributed, enforce a common geometry between the different surveys. We can evaluate the number of common tie points between surveys by comparing the number of points in each sparse cloud following chunk duplication and photo removal (Figure 2) to the number of points in the sparse cloud generated during the combined

140   alignment. Tie points generated using only photos from survey 1 will be removed when the photos from survey 1 are removed, while tie points generated using photos from both surveys will remain. If common tie points were generated, the two separated sparse clouds have more total points than the original, with the difference being the number of common points (Table 1). Note that this is distinct from the number of matches, as each tie point may be used in multiple matches.

Even when relatively few common tie points are generated, or when they are irregularly distributed, a successful alignment

145   can be achieved. For example, Figure 5 shows a section of the Rügen study area that is heavily vegetated, with only isolated patches of bare cliffs. While no common tie points can be generated in the vegetated areas, as long as there are common tie points distributed throughout the cliff sections, a relatively good comparative accuracy can be achieved, as illustrated for April 2018 – May 2018 (Figure 5B). However, if there are sections of the cliff where no matches can be made, then large comparative errors can result, as is shown in Figure 5C for the survey pair Oct. 2017 – April 2018. This survey pair had both

150   a low number (1355) and percentage (0.4%) of common tie points compared to the April 2018 – May 2018 pair, which had 3402, or 1% common tie points. More importantly, there were no common tie points generated in a ~350 m long stretch at one end of the model, leading to up to 1.5 m of comparative error in this section of the cliff. This illustrates that if common tie points are not distributed through the full extent of the model, edges of the models may not align well. The Daan River example further demonstrates that the distribution of tie points is more important than their number, as good alignment was

155   achieved throughout most of the model despite the generation of only 900 common tie points (0.3% of the total).

**4.1 Potential limitations**

In order to get a successful alignment, tie points linking the photos from different surveys must be detected. If the appearance of the area changes too much between surveys or if too much of the area of interest has changed, sufficient tie

**Página: 5**

I recommend discussing in more detail each of the results in Fig. 5 and 6.

This table must be improved: using capital letters at the beginning of the column titles, better structure, etc... The references to the study sites are confusing, please include alwasy the main name of the site and then the particular name of the area. Why not being consistent with Daan river results first and then Rugen? It is confusing.

I think there are two spaces here.

[revised manuscript text omitted]

---

## Author Comment (AC1) · 13 Aug 2019

We greatly appreciate the two helpful reviews, and will use the comments to improve the manuscript. Below, we provide a detailed response to the reviewer comments, outlining how we will revise the manuscript. We have responded to each comment (except for some very minor ones regarding typos), with the referee comments in blue italics and our responses in normal text.

**Referee 1**

*The short communication introduces a workflow to process multi-temporal data for accurate change detection although no GCPs are available. Thereby, images from multiple survey campaigns are processed at once. Afterwards, the orientated images of the individual surveys are split to retrieve the corresponding point clouds of each campaign for change detection. The idea is simple but very effective. The manuscript is well-structured and easy to follow. The results are presented comprehensively and support the introduced method. However, some issues remain regarding the explanation of the approach (especially terminology) that should be addressed in a revised manuscript. Furthermore, the authors should consider the Time-SIFT publication by Feurer & Vinatier (2018) because it describes a similar approach more detailed for applications to archival imagery. Please, see below for some detailed comments.*

Thanks a lot for pointing us towards the Feurer and Vinatier paper. We had not seen this, and it is indeed quite similar to our approach. It is a little bit of a relief to see that we are not the only ones to have thought of this nontraditional approach. We will certainly cite it and add it to our discussion of previous work.

*P1L23: It is not clear what the authors refer to with camera optical parameters. Are these the interior orientation parameters. If yes, it should be mentioned that the GCPs are also used to refine the parameters of the exterior orientation besides the interior parameters.*

Yes, this was unclear. We will restate that GCPs are used to georeference the model and to improve the calculation of both camera interior parameters and camera positions and orientations.

*P1L29: It might be better to refer to dGNSS instead of dGPS as also other satellites can be used for geo-referencing.*

Good point, will be changed.

*P2L33-35: Are these control points tie points or ground control points? If they are GCPs, where does the reliable/accurate 3D information come from? And if they are tie points, I would avoid the term control points.*

Peppa et al., 2019 refer to them as pseudo-GCPs. We will use that instead of the term control points.

*P3L65-71: This paragraph seems to be a little bit off-topic if it is left as it is. A better explanation why these challenges are displayed should be provided. For instance, why is the changing appearance of the cliff relevant? Does that potentially impact feature detection and matching? Furthermore, a final statement might improve that paragraph, as well, highlighting that this study at the cliff is a very suitable study to demonstrate the usability/necessity/benefits of the authors' approach. Although, this intention is probably meant in the paragraph it might be suitable to mention this explicitly.*

Good point. We will remove the part about the changing appearance, as it's not really relevant. Then we will explicitly state that these challenges are the reason why this is a good test case.

*P3 chapter Methods: I would suggest to include sub-headings for data acquisition and co-alignment processing to improve the readability.*

These will be added

*P3L75-76: Did the authors also consider check points as an independent reference of the reconstruction accuracy? With 14 and 12 GCPs this should be possible.*

We did not do this, as we rely on the accuracy study in Cook, 2017, which was conducted at the same site using the same control points, to estimate uncertainties (as stated in the text).

*P3L94: I thought, only the Mavic Pro was used for data acquisition (but also a Phantom is mentioned here)?*

The Phantom 3 was used for the Daan River surveys (this was mentioned in line 73).

*P4L97: What is the unit for the reconstruction uncertainty? According to Agisoft, the reconstruction uncertainty somehow relates to the base-height ratio. But how is the reconstruction uncertainty calculated?*

This value has no unit, as it is the ratio between the variation in the direction of maximum variation and the variation in the direction of minimum variation. We are reporting the parameters used in Photoscan for completeness, not because they are particularly important for the method. For the case of surveys with a lot of oblique photos, we have found that filtering by reconstruction uncertainty is the best way to remove tie points that are clearly erroneous. Other users may clean and optimize the survey differently; it doesn't really matter in terms of the co-alignment method. Because this is not a particularly important step, we don't think it's necessary to give an introduction to how Photoscan calculates it.

*P4L98: How is the adaptive camera model fitting working? What is the difference to the approach without adaptive fitting?*

As with the comment above, we are not sure that this manuscript is the place to explain the details of Photoscan's methods, but can provide a reference to the user manual.

*P4L99: The fine registration in CloudCompare is done via ICP (iterative closest point) fitting. Maybe, it might be preferable to state the actual performed algorithm rather than the tool name.*

This will be changed.

*P4L105: The alignment optimization is actually also a bundle adjustment, however considering some refined parameter settings and/or referencing information. Thus, this might be rephrased to avoid confusion of the reader.*

This will be rephrased.

*P4L116-118: Is it possible to express these differences between both change maps in numbers, e.g. considering the average of deviations between both maps? This question would also be relevant for the*

*Rügen analysis. Furthermore, did the authors also check accuracies at check points? They might be helpful to assess how well changes are detectable with the reference in general.*

We can provide the average change for each map, but we feel that the histograms shown in the figures are more informative. One issue is that some of the differences calculated are real, so a smaller average difference does not necessarily mean a better result. Unfortunately, we cannot directly compare the two Daan River change maps because the models without GCPs are warped relative to the models with GCPs, so the change maps don't align with each other.

For Rügen, as mentioned in the text, we have no independently measured check points. If we had the ability to have such points, then we would also be able to use GCPs and wouldn't have the need for this workflow. At the Daan River site, we rely on the accuracy study in Cook, 2017, which was conducted at the same site using the same control points, to estimate uncertainties.

*P4L124-125: However, this depends on how the models are aligned. If GCPs or stable areas are used, I am not certain if this statement still holds. Of course, if ICP is used than these distortions can lead to difficulties in the alignment (depending on how strong these distortions are).*

If alignment involves just rotation and transformation, then distortions will prevent good alignment of the whole model no matter what method is used. Perhaps alignment is not the best term to use here, as we are talking about only transformation and rotation of dense point clouds or meshes; we can see how this can be confused with camera alignment. We can substitute co-registration for alignment.

*P5L128-129: I am not sure if I understand that sentence correctly. Changes between 1 and 2 m are common at the observed cliff on Rügen? Thus, the noise in the data is higher than the common changes at the cliff?*

We will rephrase this. In this comparison, up 1-2 meters of change are erroneously detected in many stable areas, indicating that real changes of this magnitude would be below the level of detection.

*L123-129: Maybe the entire paragraph can be rewritten to improve clarity regarding model related distortions and issues due to alignment approaches.*

Hopefully this will be more clear by using "co-registration" rather than "alignment." But basically, if models are distorted relative to each other, then they will not perfectly fit together no matter what method you use.

*P5L130-131: Maybe it is worth to extent the explanation that the simultaneous alignment of all campaigns leads to the circumstance that the highly spatially correlated errors (James et al., 2017), which also depend on the image observations (i.e. tie points), are potentially situated at the same locations in the individual models (because image orientation across surveys are constrained to the same tie-points) and therefore mitigated during point cloud differencing.*

Yes, this is exactly what we are trying to convey – that the models contain errors, but they are consistent across the different surveys, so they don't influence the change detection. We will add a sentence to say this explicitly.

*P5L153: I would not state that edges are the issue but rather areas outside the tie point region.*

In these surveys, it does seem to be more an issue of edges, as the extents of the dense and sparse clouds are the same. The points near the edge are generally only seen on two or three photos, and they are only seen on the same edge of these photos, so they are less well-constrained (compared to points which are visible on many photos and which are located at a range of positions on different photos).

*P6L160-163: Maybe this statement should be separated more clearly from the previous because another aspect is discussed. The first aspect is referring to too strong changes of the surface and therefore failing to find matches and the second refers to changes of the entire surface but remaining a general similar appearance and thus falsely retrieving matches.*

Yes, we can clarify this. The method requires matches (the first aspect), and it requires accurate matches (the second aspect). We can't comment too much on false matches because we didn't find any, but we raise this as something to watch out for.

*P6L164: What do the authors refer to when they are talking about scaling between numbers of photos?*

This refers to the nonlinear increase in processing time when more photos are added – doubling the photos will more than double the time required for point matching and camera alignment.

*P6L164-168: I have a little bit difficulty to understand that sentence. Do the authors mean that with each new campaign all the campaigns have to be re-processed?*

Yes, this is what we mean. Any previous campaigns that you would like to compare to the new campaign must be reprocessed.

*P6L168-169: Might it not be possible to only compare from one survey to the next to avoid increasing the processing time with each new survey, although this might be less favorable for error propagation? Maybe it might worth testing in a future study how well campaign to campaign processing performs compared to reprocessing everything.*

Of course, and this is what we typically do with the Rügen surveys. But this means that for four surveys A, B, C, and D, you will need to do all of the processing three times – A+B, B+C, and C+D. So if it takes 10 hours to generate the dense cloud for survey A, you will have to do that twice. And you can't compare A vs. C or B vs. D. Basically, the issue is that when you conduct a new survey, you can't re-use any of the models you have previously generated to compare with it. This is quite different from the typical workflow, where you create a "final" model that can be compared to any future models.

*P6L177-178: Maybe the combination of both is most suitable (e.g. as discussed by Feurer & Vinatier, 2018). Align all campaigns in one workflow (this might also improve general model accuracy as more image observations will be available) and scale/georeference the whole project with GCPs (from just one campaign).*

This is a great suggestion, thanks. We will add a sentence about this, with a citation to Feurer and Vinatier.

**Referee 2**

*This is an interesting study on the possibility of improving the comparative accuracy of multiple surveys by co-processing the image sets when stable areas can be found and matched in a particular area. Rather than the workflow itself (which is hardly a proper workflow but just a modification of the standard SfM pipeline), I found the greatest merit of this work drawing the attention to this co-alignment possibility, that in many cases my be discarded or overlooked and can help to improve the quality of the results. My main comments to this work are the following (please check the annotated pdf for specific comments throughout the manuscript): 1. I would strongly suggest to include in the manuscript title the main limitation of the workflow. i.e. the presence of stable areas. The authors have acknowledged this in the limitations and conclusions sections and should be specified in the title since is a major requirement. 2. The authors are too focused on the geomorphological settings (cliffs, rivers and such), which is not bad, but a relevant part of the SfM community works on more artificial environments such as agricultural settings where hardly stable areas can be found. How applicable would be the co-aligment in these cases? A quick literature review could give the authors a general view of the types of scenarios in which the SfM approaches are being applied and maybe they could comment in more detail to what extent their method is feasible to be applied.*

The limitations of the method are made quite clear in the manuscript; we disagree that it's necessary to include mention of stable areas in the title and feel that it would make the title long, awkward, and too detailed. We will add a sentence to the abstract specifying that the method relies of the presence of stable areas.

The goal of this manuscript to introduce a solution for people working in environments where ground control is impossible or very difficult to obtain. Thus, we are not focused on places like agricultural settings or other artificial environments, as traditional methods work just fine in these areas. Plus, since we have no experience with or data from such areas, we have no idea how well the method will perform. It may be that peripheral stable infrastructure such as buildings, roads, fences, etc. may provide sufficient common tie points, and maybe not. It may also vary depending on the location. We hope that people working in different settings can give it a try and evaluate whether it works for their particular area.

*Número: 5 Autor: Asunto: Resaltado Fecha: 17/07/2019 7:31:01 1. Please justify the selection of these two study areas 2. Rephrase the sentence to be shorter and better structured*

We will rephrase this. The appropriateness of the study areas are justified in more detail in the area descriptions below.

*Número: 6 Autor: Asunto: Resaltado Fecha: 17/07/2019 7:30:56 Include company and country or reference.*

Agisoft is the company, so that is already there. We have never seen the country listed in other publications that use Photoscan. We did neglect to provide the version number, so we will add this.

*Número: 8 Autor: Asunto: Resaltado Fecha: 17/07/2019 7:38:04 No information of this study area has been included as a figure. Please provide a map and picture if appropriate, similarly to the Rugen site.*

We felt that this information wasn't necessary, as it is available in a previous publication and isn't critical for interpreting the results.

*Número: 9 Autor: Asunto: Resaltado Fecha: 17/07/2019 7:33:21 Autor: Asunto: Nota adhesiva Fecha: 17/07/2019 7:33:40 Why not describing this first, being the primary area?*

We use the Daan case as a kind of proof of concept, where we apply the method to a more traditional type of survey in an area where traditional methods are possible, so we present it first. We describe it first because we present it first in the results. We consider Rügen to be the primary area, as it is the setting where traditional methods can't be used and our co-registration workflow is really necessary to get useful change detection results.

*This was made after or before the fine registration and M3C2 algorithm?*

This was done after the M3C2 calculations – the steps were done in the order that they are presented in the text.

*Número: 2 Autor: Asunto: Resaltado Fecha: 17/07/2019 7:47:23 Can you provide the results when both surveys are processed independently (no co-alignment)?*

We will add this analysis to figure 3.

*Página: 5 Número: 1 Autor: Asunto: Resaltado Fecha: 17/07/2019 7:56:33 I recommend discussing in more detail each of the results in Fig. 5 and 6.*

We are not sure what additional detail is expected. To discuss the patterns of cliff collapse and retreat on Rügen is far beyond the scope of this manuscript, particularly since it is a short communication and not a full research paper.

*Número: 2 Autor: Asunto: Resaltado Fecha: 17/07/2019 8:00:42 This table must be improved: using capital letters at the beginning of the column titles, better structure, etc... The references to the study sites are confusing, please include always the main name of the site and then the particular name of the area. Why not being consistent with Daan river results first and then Rugen? It is confusing.*

We will reorganize the table and clarify the study site names.

*The examples provided by the authors are typically focused on geomorphological settings which include stable areas. What about other scenarios such as agricultural settings where SfM is being frequently used? I would recommend revising other settings in literature where UAV SfM is being extensively used and comment the feasibility of the workflow accordingly.*

As we say above, we don't have the data or experience to evaluate the feasibility of the approach in such settings, and it may vary depending on the details of the agricultural setting and the surveys. Such a discussion would only be speculation, which we don't feel is very useful. We are open about what is required to make the approach work, and we hope that readers can evaluate their own areas on the basis of that.

*Número: 2 Autor: Asunto: Resaltado Fecha: 17/07/2019 8:03:23 What does this mean? Different surveys may have different target accuracies depending on the aims of the study*

Survey-grade accuracy is a term that is commonly used in the surveying community and typically refers to accuracy on the order of cm or better.

---

## Author Response (AR1)

We greatly appreciate the two helpful reviews, and have used the comments to improve the manuscript. Below, we provide a detailed response to the reviewer comments, outlining how we have revised the manuscript. We have responded to each comment (except for some very minor ones regarding typos), with the referee comments in blue italics and our responses in normal text.

**Referee 1**

*The short communication introduces a workflow to process multi-temporal data for accurate change detection although no GCPs are available. Thereby, images from multiple survey campaigns are processed at once. Afterwards, the orientated images of the individual surveys are split to retrieve the corresponding point clouds of each campaign for change detection. The idea is simple but very effective. The manuscript is well-structured and easy to follow. The results are presented comprehensively and support the introduced method. However, some issues remain regarding the explanation of the approach (especially terminology) that should be addressed in a revised manuscript. Furthermore, the authors should consider the Time-SIFT publication by Feurer & Vinatier (2018) because it describes a similar approach more detailed for applications to archival imagery. Please, see below for some detailed comments.*

Thanks a lot for pointing us towards the Feurer and Vinatier paper. We had not seen this, and it is indeed quite similar to our approach. It is a little bit of a relief to see that we are not the only ones to have thought of this nontraditional approach. We have added it to our discussion of previous work and have specified that our proposed method is a generalization of this one.

*P1L23: It is not clear what the authors refer to with camera optical parameters. Are these the interior orientation parameters. If yes, it should be mentioned that the GCPs are also used to refine the parameters of the exterior orientation besides the interior parameters.*

Yes, this was unclear. We restated that GCPs are used to georeference the model and to improve the calculation of both camera interior parameters and camera positions and orientations.

*P1L29: It might be better to refer to dGNSS instead of dGPS as also other satellites can be used for geo-referencing.*

Good point, this was changed.

*P2L33-35: Are these control points tie points or ground control points? If they are GCPs, where does the reliable/accurate 3D information come from? And if they are tie points, I would avoid the term control points.*

Peppa et al., 2019 refer to them as pseudo-GCPs. We will use that instead of the term control points.

*P3L65-71: This paragraph seems to be a little bit off-topic if it is left as it is. A better explanation why these challenges are displayed should be provided. For instance, why is the changing appearance of the cliff relevant? Does that potentially impact feature detection and matching? Furthermore, a final statement might improve that paragraph, as well, highlighting that this study at the cliff is a very suitable study to demonstrate the usability/necessity/benefits of the authors' approach. Although, this intention is probably meant in the paragraph it might be suitable to mention this explicitly.*

Good point. We have removed the part about the changing appearance, as it's not really relevant. We now explicitly state that these challenges are the reason why this is a good test case.

*P3 chapter Methods: I would suggest to include sub-headings for data acquisition and co-alignment processing to improve the readability.*

These have been added.

*P3L75-76: Did the authors also consider check points as an independent reference of the reconstruction accuracy? With 14 and 12 GCPs this should be possible.*

We did not do this, as we rely on the accuracy study in Cook, 2017, which was conducted at the same site using the same control points, to estimate uncertainties (as stated in the text).

*P3L94: I thought, only the Mavic Pro was used for data acquisition (but also a Phantom is mentioned here)?*

The Phantom 3 was used for the Daan River surveys (this was mentioned in line 73).

*P4L97: What is the unit for the reconstruction uncertainty? According to Agisoft, the reconstruction uncertainty somehow relates to the base-height ratio. But how is the reconstruction uncertainty calculated?*

This value has no unit, as it is the ratio between the variation in the direction of maximum variation and the variation in the direction of minimum variation. We are reporting the parameters used in Photoscan for completeness, not because they are particularly important for the method. For the case of surveys with a lot of oblique photos, we have found that filtering by reconstruction uncertainty is the best way to remove tie points that are clearly erroneous. Other users may clean and optimize the survey differently; it doesn't really matter in terms of the co-alignment method. Because this is not a particularly important step, we don't think it's necessary to give an introduction to how Photoscan calculates it.

*P4L98: How is the adaptive camera model fitting working? What is the difference to the approach without adaptive fitting?*

As with the comment above, we are not sure that this manuscript is the place to explain the details of Photoscan's methods.

*P4L99: The fine registration in CloudCompare is done via ICP (iterative closest point) fitting. Maybe, it might be preferable to state the actual performed algorithm rather than the tool name.*

This was changed.

*P4L105: The alignment optimization is actually also a bundle adjustment, however considering some refined parameter settings and/or referencing information. Thus, this might be rephrased to avoid confusion of the reader.*

This was rephrased to: "the point detection and matching, initial bundle adjustment, and optimization"

*P4L116-118: Is it possible to express these differences between both change maps in numbers, e.g. considering the average of deviations between both maps? This question would also be relevant for the*

*Rügen analysis. Furthermore, did the authors also check accuracies at check points? They might be helpful to assess how well changes are detectable with the reference in general.*

We can provide the average change for each map, but we feel that the histograms shown in the figures are more informative. One issue is that some of the differences calculated are real, so a smaller average difference does not necessarily mean a better result. Unfortunately, we cannot directly compare the two Daan River change maps because the models without GCPs are warped relative to the models with GCPs, so the change maps don't align with each other.

For Rügen, as mentioned in the text, we have no independently measured check points. If we had the ability to have such points, then we would also be able to use GCPs and wouldn't have the need for this workflow. At the Daan River site, we rely on the accuracy study in Cook, 2017, which was conducted at the same site using the same control points, to estimate uncertainties.

*P4L124-125: However, this depends on how the models are aligned. If GCPs or stable areas are used, I am not certain if this statement still holds. Of course, if ICP is used than these distortions can lead to difficulties in the alignment (depending on how strong these distortions are).*

If alignment involves just rotation and transformation, then distortions will prevent good alignment of the whole model no matter what method is used. Perhaps alignment is not the best term to use here, as we are talking about only transformation and rotation of dense point clouds or meshes; we can see how this can be confused with camera alignment. We have substituted co-registration for alignment.

*P5L128-129: I am not sure if I understand that sentence correctly. Changes between 1 and 2 m are common at the observed cliff on Rügen? Thus, the noise in the data is higher than the common changes at the cliff?*

We have rephrased this to: "Throughout the model area, up 1-2 meters of change are erroneously detected in many stable areas, indicating that real changes of this magnitude would be below the level of detection. For the Rügen study area, this level of detection would preclude the use of UAV surveys to monitor small cliff failures."

*L123-129: Maybe the entire paragraph can be rewritten to improve clarity regarding model related distortions and issues due to alignment approaches.*

Hopefully this will be more clear by using "co-registration" rather than "alignment." But basically, if models are distorted relative to each other, then they will not perfectly fit together no matter what method you use, but you might fit them together badly in different ways.

*P5L130-131: Maybe it is worth to extent the explanation that the simultaneous alignment of all campaigns leads to the circumstance that the highly spatially correlated errors (James et al., 2017), which also depend on the image observations (i.e. tie points), are potentially situated at the same locations in the individual models (because image orientation across surveys are constrained to the same tie-points) and therefore mitigated during point cloud differencing.*

Yes, this is exactly what we are trying to convey – that the models contain errors, but they are consistent across the different surveys, so they don't influence the change detection. We have rephrased as: "When the cameras from multiple surveys are co-aligned, the resulting point clouds still contain distortions, but if the procedure is successful, they have been fit to a common geometry and the

distortions are consistent between the models. As a result, these errors do not influence comparisons between the models, comparative accuracy is much higher and robust change detection can be performed."

*P5L153: I would not state that edges are the issue but rather areas outside the tie point region.*

In these surveys, it does seem to be more an issue of edges, as the extents of the dense and sparse clouds are the same. The points near the edge are generally only seen on two or three photos, and they are only seen on the same edge of these photos, so they are less well-constrained (compared to points which are visible on many photos and which are located at a range of positions on different photos).

*P6L160-163: Maybe this statement should be separated more clearly from the previous because another aspect is discussed. The first aspect is referring to too strong changes of the surface and therefore failing to find matches and the second refers to changes of the entire surface but remaining a general similar appearance and thus falsely retrieving matches.*

We have clarified this and modified this paragraph to:

"In order to get a successful alignment, tie points linking the photos from different surveys must be detected, and false matches must be avoided. If the appearance of the area changes too much between surveys or if too much of the area of interest has changed, sufficient tie points may not be generated, as described above. Therefore, well-distributed stable areas with a consistent appearance are required for successful alignment. In the examples presented here, we did not observe any false matches, as surface changes were always accompanied by changes in appearance, preventing the detection of matches in unstable areas. In settings with large-scale surface deformation, such as a slow moving landslide or deep seated gravitational slope deformation, this may not be the case, and it is possible that points may be matched in unstable areas. In such settings, care should be taken to evaluate the reliability of the common tie points."

*P6L164: What do the authors refer to when they are talking about scaling between numbers of photos?*

This refers to the nonlinear increase in processing time when more photos are added – doubling the photos will more than double the time required for point matching and camera alignment. We have modified to "Due to non-linear scaling between the number of photos and the processing time"

*P6L164-168: I have a little bit difficulty to understand that sentence. Do the authors mean that with each new campaign all the campaigns have to be re-processed?*

Yes, this is what we mean. Any previous campaigns that you would like to compare to the new campaign must be reprocessed.

*P6L168-169: Might it not be possible to only compare from one survey to the next to avoid increasing the processing time with each new survey, although this might be less favorable for error propagation? Maybe it might worth testing in a future study how well campaign to campaign processing performs compared to reprocessing everything.*

Of course, and this is what we typically do with the Rügen surveys. But this means that for four surveys A, B, C, and D, you will need to do all of the processing three times – A+B, B+C, and C+D. So if it takes 10 hours to generate the dense cloud for survey A, you will have to do that twice. And you can't compare A

vs. C or B vs. D. Basically, the issue is that when you conduct a new survey, you can't re-use any of the models you have previously generated to compare with it. This is quite different from the typical workflow, where you create a "final" model that can be compared to any future models. This paragraph has been modified to: "For a single pair of surveys, the co-alignment workflow has a limited impact on processing time. Due to non-linear scaling between the number of photos and the processing time, performing the point matching and camera alignment step once with n photos will take longer than performing it twice with n/2 photos, but this effect will be relatively minor until the number of photos becomes large. The more significant impact on processing time comes from the requirement that for each survey set to be compared, the entire chain of processing from point matching to dense cloud construction must be redone. This can greatly increase the total processing time for large sets of surveys. For example, for a set of four surveys, A, B, C, and D, a series of pairwise processing and comparison (A-B, B-C, C-D) would require the point matching and camera alignment step to be performed three times and would require the construction of six dense clouds (surveys B and C would each have two dense clouds). This processing time can be reduced by applying the method to larger sets of surveys. We have simultaneously co-aligned photographs from up to 4 different epochs to obtain a set of mutually comparable point clouds from 2017-2018 (Figure 6). In some cases, an unsuccessful alignment of two surveys can be improved by adding a third survey. For example, if changes in surface appearance (lighting, shadows) or in camera obliquity prevent the detection of sufficient common tie points between the original two surveys, a third survey that generates enough common tie points with each of the original two can lead to successful alignment of all three surveys. However, despite the possibilities for batch processing, the fundamental drawback of this method is that it does not result in a definitive model for a given survey period - models that were constructed based on co-alignment of one set of surveys cannot be re-used for comparison to an additional survey."

*P6L177-178: Maybe the combination of both is most suitable (e.g. as discussed by Feurer & Vinatier, 2018). Align all campaigns in one workflow (this might also improve general model accuracy as more image observations will be available) and scale/georeference the whole project with GCPs (from just one campaign).*

This is a great suggestion, thanks. We have added a sentence about this, with credit to Feurer and Vinatier.

**Referee 2**

*This is an interesting study on the possibility of improving the comparative accuracy of multiple surveys by co-processing the image sets when stable areas can be found and matched in a particular area. Rather than the workflow itself (which is hardly a proper workflow but just a modification of the standard SfM pipeline), I found the greatest merit of this work drawing the attention to this co-alignment possibility, that in many cases my be discarded or overlooked and can help to improve the quality of the results. My main comments to this work are the following (please check the annotated pdf for specific comments throughout the manuscript): 1. I would strongly suggest to include in the manuscript title the*

*main limitation of the workflow. i.e. the presence of stable areas. The authors have acknowledged this in the limitations and conclusions sections and should be specified in the title since is a major requirement. 2. The authors are too focused on the geomorphological settings (cliffs, rivers and such), which is not bad, but a relevant part of the SfM community works on more artificial environments such as agricultural settings where hardly stable areas can be found. How applicable would be the co-aligment in these cases? A quick literature review could give the authors a general view of the types of scenarios in which the SfM approaches are being applied and maybe they could comment in more detail to what extent their method is feasible to be applied.*

The limitations of the method are made quite clear in the manuscript; we disagree that it's necessary to include mention of stable areas in the title and feel that it would make the title long, awkward, and too detailed. We added a sentence to the abstract specifying that the method relies of the presence of stable areas: "The method is based on the automated detection of common tie points in stable portions of the survey area."

The goal of this manuscript to introduce a solution for people working in environments where ground control is impossible or very difficult to obtain. Thus, we are not focused on places like agricultural settings or other artificial environments, as traditional methods work just fine in these areas. Plus, since we have no experience with or data from such areas, we have no idea how well the method will perform. It may be that peripheral stable infrastructure such as buildings, roads, fences, etc. may provide sufficient common tie points, and maybe not. It may also vary depending on the location. We hope that people working in different settings can give it a try and evaluate whether it works for their particular area.

*Número: 5 Autor: Asunto: Resaltado Fecha: 17/07/2019 7:31:01 1. Please justify the selection of these two study areas 2. Rephrase the sentence to be shorter and better structured*

We have rephrased this sentence to: "Here, we introduce a simple workflow involving the co-alignment of photographs from different surveys; our method is similar to that of Feurer and Vinatier (2018), but uses no GCPs and is generalized to any set of repeat SfM surveys. Using data from two contrasting study areas: a bedrock gorge in Taiwan and a steep cliff coast in northern Germany, we demonstrate that we can achieve high comparative survey accuracy and low limits of change detection using a low-cost off the shelf UAV without ground control points."

The appropriateness of the study areas are justified in the area descriptions below; we don't think this level of detail is needed in the introduction.

*Número: 6 Autor: Asunto: Resaltado Fecha: 17/07/2019 7:30:56 Include company and country or reference.*

Agisoft is the company, so that is already there. We have never seen the country listed in other publications that use Photoscan. We did neglect to provide the version number, so we have added this.

*Número: 8 Autor: Asunto: Resaltado Fecha: 17/07/2019 7:38:04 No information of this study area has been included as a figure. Please provide a map and picture if appropriate, similarly to the Rugen site.*

We felt that this information wasn't necessary, as it is available in a previous publication and isn't critical for interpreting the results.

*Número: 9 Autor: Asunto: Resaltado Fecha: 17/07/2019 7:33:21 Autor: Asunto: Nota adhesiva Fecha: 17/07/2019 7:33:40 Why not describing this first, being the primary area?*

We use the Daan case as a kind of proof of concept, where we apply the method to a more traditional type of survey in an area where traditional methods are possible, so we present it first. We describe it first because we present it first in the results. We consider Rügen to be the primary area, as it is the setting where traditional methods can't be used and our co-registration workflow is really necessary to get useful change detection results.

*This was made after or before the fine registration and M3C2 algorithm?*

This was done after the M3C2 calculations – the steps were done in the order that they are presented in the text. We have clarified this.

*Número: 2 Autor: Asunto: Resaltado Fecha: 17/07/2019 7:47:23 Can you provide the results when both surveys are processed independently (no co-alignment)?*

We have added this result to figure 3.

*Página: 5 Número: 1 Autor: Asunto: Resaltado Fecha: 17/07/2019 7:56:33 I recommend discussing in more detail each of the results in Fig. 5 and 6.*

We are not sure what additional detail is expected. To discuss the patterns of cliff collapse and retreat on Rügen is far beyond the scope of this manuscript, particularly since it is a short communication and not a full research paper.

*Número: 2 Autor: Asunto: Resaltado Fecha: 17/07/2019 8:00:42 This table must be improved: using capital letters at the beginning of the column titles, better structure, etc... The references to the study sites are confusing, please include always the main name of the site and then the particular name of the area. Why not being consistent with Daan river results first and then Rugen? It is confusing.*

We have reorganized the table and clarified the study site names.

*The examples provided by the authors are typically focused on geomorphological settings which include stable areas. What about other scenarios such as agricultural settings where SfM is being frequently used? I would recommend revising other settings in literature where UAV SfM is being extensively used and comment the feasibility of the workflow accordingly.*

As we say above, we don't have the data or experience to evaluate the feasibility of the approach in such settings, and it may vary depending on the details of the agricultural setting and the surveys. Such a discussion would only be speculation, which we don't feel is very useful. We are open about what is required to make the approach work, and we hope that readers can evaluate their own areas on the basis of that.

*Número: 2 Autor: Asunto: Resaltado Fecha: 17/07/2019 8:03:23 What does this mean? Different surveys may have different target accuracies depending on the aims of the study*

Survey-grade accuracy is a term that is commonly used in the surveying community and typically refers to 
[revised manuscript text omitted]

[Figure]

Jan 2018 point cloud

A

May 2017 - Jan 2018
differences with GCPs

B

May 2017 - Jan 2018
differences with no GCPs

C

100 m

M3C2 change (meters)

-0.5    -0.2  -0.1   0   0.1  0.2      0.5

[Figure]

Figure 3: Daan River comparisons. A) Jan. 2018 point cloud with the ground control points shown. B) M3C2 differences between May 2017 and Jan. 2018 point clouds  processed separately with no GCPs. C) M3C2 differences between May 2017 and Jan. 2018 point clouds processed using the co-alignment workflow. D) M3C2 differences between May 2017 and Jan. 2018 point clouds processed separately using GCPs. E) Density curves of the measured changes shown in B , C, and D.

290

[Figure]

Figure 4: Cloud-cloud differences between the April 2018 and May 2018 surveys in the Kieler Bach section of the coast, calculated using the M3C2 algorithm. A) April 2018 point cloud. B) May 2018 point cloud. C) M3C2 differences between point clouds created

295 using the standard workflow. D) M3C2 differences between point clouds created using the co-alignment workflow. High values of positive change at the top of the cliff are due to leaf growth on the trees. Isolated sections of positive change on the cliff face are also related to growth of bushes and trees. In panel D, several small failure events can be identified on the cliff face (circled). These have been confirmed visually using the before and after photographs. E) HistogramsDensity curves of the measured changes shown in C and D.

300

[Figure]

**Figure 5: Cloud-cloud differences in the heavily vegetated Königsstuhl section of the coast. A) May 2018 point cloud showing the extent of the vegetation. B) M3C2 differences between April 2018 and May 2018 point clouds. The vegetation has been removed using standard deviation and point density filters. Leaf growth results in very high measured changes in the vegetated areas, so only the bedrock cliff sections are shown. C) M3C2 differences between Oct. 2017 and April 2018 point clouds. A lack of common tie points detected in the left side of the region results in relative distortion of the models and high errors in the change detection.**

305

310

[Figure]

**Figure 6: Changes calculated from batch co-alignment of four surveys simultaneously. A-C) M3C2 changes between successive surveys following simultaneous co-alignment. In panel B) bands of change in the lower half of the cliff show more diffuse erosion**
315    **due to mechanical weathering. D-F) the same comparisons following the separate processing workflow.**

| Survey | UAV | Number of photographs | Sparse cloud points | Common tie points | % Common tie points |
|---|---|---|---|---|---|
| **Daan River (figure 3)** | | | | | |
| May 17, 2017 | Phantom 3 Adv. | 197 | 136479 | - | - |
| Jan. 30, 2018 | Phantom 3 Adv. | 298 | 168953 | | - |
| Combined alignment | | - | 304532 | 900 | 0.30 |
| - | | | | - | |
| **Rügen Kieler Bach (figure 4)** | | | | - | |
| April 03, 2018 | Mavic Pro | 442 | 125634 | | |
| May 29, 2018 | Mavic Pro | 331 | 200863 | | |
| Combined alignment | | | 313513 | 12984 | 4.14 |
| | | | | | |
| **** | | | | - | |
|  |  |  |  | - | - |
|  |  |  |  | | - |
|  |  |  |  | | - |
|  |  |  |  | | - |
|  | | - |  |  |  |
| - | | | | - | |
| - | | | | - | |
| **Rügen Konigsstuhl (figure 5)** | | | | | |
| April 03, 2018 | Mavic Pro | 250 | 111464 | | |
| May 29, 2018 | Mavic Pro | 249 | 157677 | | |
| Combined alignment | | | 264597 | 4544 | 1.72 |
| | | | | | |
| Oct. 18, 2017 | Mavic Pro | 414 | 195901 | | |
| April 03, 2018 | Mavic Pro | 246 | 117227 | | |
| Combined alignment | | | 311773 | 1355 | 0.43 |
| | | | | | |
| **Rügen batch processing (figure 6)** | | | | | |
| **** | | | | - | |
| Oct. 18, 2017 | Mavic Pro | 839 | 363485 | | |
| Jan. 24, 2018 | Mavic Pro | 338 | 125121 | | |
| April 03, 2018 | Mavic Pro | 442 | 128640 | - | |
| May 29, 2018 | Mavic Pro | 575 | 324391 | - | |

| Combined alignment | 918741 | 22896 | 2.49 |

320 **Table 1: survey characteristics**

---

## Author Response (AR2)

We have gone through the manuscript and carefully checked the writing for repetition and other minor issues.